# Activation of Non-Canonical Autophagic Pathway through Inhibition of Non-Integrin Laminin Receptor in Neuronal Cells

**DOI:** 10.3390/cells11030466

**Published:** 2022-01-29

**Authors:** Adriana Limone, Iolanda Veneruso, Antonella Izzo, Maurizio Renna, Raffaella Bonavita, Silvia Piscitelli, Gaetano Calì, Sergio De Nicola, Patrizia Riccio, Valeria D’Argenio, Antonio Lavecchia, Daniela Sarnataro

**Affiliations:** 1Department of Molecular Medicine and Medical Biotechnology, University of Naples “Federico II”, Via S. Pansini 5, 80131 Naples, Italy; adriana.limone@unina.it (A.L.); venerusoi@ceinge.unina.it (I.V.); antonella.izzo@unina.it (A.I.); maurizio.renna@unina.it (M.R.); bonavitaraffaella@gmail.com (R.B.); silvia.piscitelli@unina.it (S.P.); patrizia.riccio@unina.it (P.R.); 2CEINGE-Biotecnologie Avanzate Scarl, Via G. Salvatore 486, 80145 Naples, Italy; dargenio@ceinge.unina.it; 3Institute of Endocrinology and Oncology of CNR, 80131 Naples, Italy; g.cali@ieos.cnr.it; 4Department of Physics “E. Pancini”, University of Naples “Federico II”, Complesso Universitario di M.S. Angelo, 80126 Naples, Italy; sergio.denicola@unina.it; 5CNR-SPIN Institute, National Research Council, 80126 Naples, Italy; 6Department of Human Sciences and Quality of Life Promotion, San Raffaele Open University, Via di Val Cannuta 247, 00166 Rome, Italy; 7Department of Pharmacy, “Drug Discovery Lab”, University of Naples “Federico II”, Via D. Montesano 49, 80131 Naples, Italy; antonio.lavecchia@unina.it

**Keywords:** non-canonical autophagy, m-TOR-independent autophagy, 37/67 kDa laminin receptor (LR), ribosomal protein SA (RPSA), 37/67 kDa LR inhibitor, endocytic pathway, Rab proteins, ATG proteins, LC3

## Abstract

To fight neurodegenerative diseases, several therapeutic strategies have been proposed that, to date, are either ineffective or at the early preclinical stages. Intracellular protein aggregates represent the cause of about 70% of neurodegenerative disorders, such as Alzheimer’s disease. Thus, autophagy, i.e., lysosomal degradation of macromolecules, could be employed in this context as a therapeutic strategy. Searching for a compound that stimulates this process led us to the identification of a 37/67kDa laminin receptor inhibitor, NSC48478. We have analysed the effects of this small molecule on the autophagic process in mouse neuronal cells and found that NSC48478 induces the conversion of microtubule-associated protein 1A/1B-light chain 3 (LC3-I) into the LC3-phosphatidylethanolamine conjugate (LC3-II). Interestingly, upon NSC48478 treatment, the contribution of membranes to the autophagic process derived mainly from the non-canonical m-TOR-independent endocytic pathway, involving the Rab proteins that control endocytosis and vesicle recycling. Finally, qRT-PCR analysis suggests that, while the expression of key genes linked to canonical autophagy was unchanged, the main genes related to the positive regulation of endocytosis (pinocytosis and receptor mediated), along with genes regulating vesicle fusion and autolysosomal maturation, were upregulated under NSC48478 conditions. These results strongly suggest that 37/67 kDa inhibitor could be a useful tool for future studies in pathological conditions.

## 1. Introduction

Defects in the vesicle and lysosome-mediated degradative pathway, namely autophagy, are likely to contribute to neurodegenerative processes in different diseases, including Alzheimer’s disease (AD) [1]. In recent years, several efforts have focused on the identification and employment of novel drugs capable of regulating the canonical autophagic pathway [2], which is orchestrated by the hierarchical and coordinated activity of autophagy related genes (ATG) that display high homology between the yeast and mammalian genomes [3]. The autophagic machinery, which coordinates the nucleation of the isolation membrane or phagophore, is regulated by the upstream ULK1/2 complex, whose activity is controlled by the nutrient-sensing pathways (including m-TORC1 and AMPK), and by the phosphatidylinositol-3 kinase class 3-beclin 1 complex (PI3KC3/BECN1). This complex regulates the formation of phosphatidylinositol 3-phospate (PIP3) on the membrane of growing phagophores, allowing the recruitment of downstream complexes responsible for vesicle elongation, which encloses cytoplasmic components, forming a double-membrane structure defined autophagosome that, upon fusion with lysosomes, ultimately becomes autolysosome. The maturation of phagophore into autophagosome is under the control of two ubiquitin-like conjugation systems. The first one results in the formation of ATG12-ATG5-ATG16L1 multimeric complex on the membrane of the extending phagophore, which in turn is responsible for the conjugation and correct targeting of the second ubiquitin-like molecule, namely microtubule-associated protein 1A/1B-light chain 3 (LC3-I) [4].

LC3-I protein, in its lipidated form (LC3-II), is conjugated to the membrane of forming autophagosomes, and the appearance of the LC3 puncta by microscopy, together with the intensity of the lipidated isoform on SDS-PAGE, represents a standard and reliable measure of the autophagic pathway activity [5]. Beside this, it is now well established that several ATG proteins play an unconventional role in other pathways distinct from canonical autophagy and autophagosomes biogenesis. Indeed, several ATG proteins participate in a process defined as LC3-associated phagocytosis (LAP), in which they modify the phagosomal membrane to enhance degradation of phagocytosed elements, as well as a similar LAP-like lipidation of LC3 on macroendocytic vacuoles during macropinocytosis, entosis or phagocytosis of apoptotic cells [6,7].

However, it is now clear that the membrane origins of autophagosomes may involve multiple sources (among these, plasma membrane and clathrin-dependent endocytic vesicles) [8], and although the exact molecular mechanisms characterizing non-canonical autophagy remain poorly understood, LC3 lipidation can occur independently of the upstream regulators of canonical autophagy, namely the ULK1 complex [7] or BECN1 [9,10], and its features reflect the endocytosis-autophagic network activity. The interest in the endolysosomal system has recently grown because of its identification as an “emerging hub” both in human innate immune response [11] and in the pathobiology of neurodegenerative disease, such as AD [12,13]. In the present manuscript, we performed qRT-PCR analysis and found a large cluster of endocytosis-related genes, whose expression is sustained in neuronal cells by the administration of laminin-1 non integrin receptor inhibitor NSC48478 [14]. Beside this, none of the pre-initiation complex-related genes (ULK1/PI3KC3- BECN1) were affected by inhibitor treatment. The independence from the nucleation complexes is additionally sustained by the observation that BECN1 downregulation does not hamper NSC48478 effects on the autophagic process, whereas it depends on ATG16L1 activity. Moreover, the increased induction of LC3 conjugation by the inhibitor was accompanied by the activating phosphorylation of m-TOR, which, based upon our previous observations, is likely sustained by activation of the Akt pathway [15]. Our results are reinforced by previous studies, which have shown that Ser-2448 in m-TOR is a direct target site of the Akt kinase [16,17]. Collectively, our findings of upregulation of genes related to endocytosis and vesicle trafficking/fusion, as well as to autophagic flux maturation, suggest that NSC48478 inhibitor stimulates a non-canonical m-TOR independent autophagic pathway, and that it might represent a valuable tool to be tested in pathological conditions, such as Alzheimer’s disease.

## 2. Materials and Methods

### 2.1. Reagents and Antibodies

Cell culture reagents were purchased from Gibco Laboratories (Grand Island, NY, USA). NSC48478 [1-((4-chloroanilino)methyl)2-naphthol] was identified by SB-VS; it was obtained from the NCI/DTP Open Chemical Repository (http://dtp.cancer.gov, last accessed date 20 January 2022), dissolved in dimethyl sulfoxide (DMSO), and stored at −20 °C. Rapamycin (R8781), Bafilomycin A1 (B1793) and Chloroquine diphosphate salt (C6628) were from Sigma-Aldrich (St. Louis, MO, USA). EuroGold TriFast (EMR507100) for RNA isolation was from EuroClone. Lipofectamine LTX and PLUS Reagents (15338100) for cell transfection were from Invitrogen (Molecular Probes, Eugene, OR, USA). Anti-LC3B antibody (2775), anti-m-TOR (2972), anti-ATG16L1 (8089) and anti-BECN1 (3738) were from Cell Signaling Technology (Danvers, MA, USA). Anti-phospho m-TOR (Ser-2448) (67778-1-Ig) was from Proteintech (Rosemont, PA, USA). Anti-Rab7 antibody (R8779), anti-Amyloid Precursor Protein (A8717) and anti-FLAG M2 (F9291) were from Sigma-Aldrich (St. Louis, MO, USA). Anti-Rab5A antibody (sc-309) was from Santa Cruz Biotechnology. Anti-Rab8 antibody (610844), anti-Rab27 (558532) and anti-EEA1 (610456) were from BD Transduction Laboratories. Anti-alpha Tubulin antibody (ab7291) was from Abcam (Cambridge, UK). Anti-KDEL antibody was from StressGen Biotechnologies Corp (Victoria, BC, Canada). Wheat Germ Agglutinin Alexa-555 conjugate (WGA) and Transferrin Alexa-594-conjugated (Tfr) were from Invitrogen (Molecular Probes, Eugene, OR, USA). DAPI was from Cell Signaling Technology.

### 2.2. Cell Culture and Drug Treatment

GT1 (mouse hypothalamic neuronal cell line) was cultured in Dulbecco’s Modified Eagle’s Medium (DMEM), with 4500 mg/glucose/L, 110 mg sodium pyruvate and l-glutamine (Sigma-Aldrich. St. Louis, MO, USA, code D6429) supplemented with 10% foetal bovine serum. For inhibitor NSC48478 treatment, cells were washed in serum-free media and incubated for 24 h at 37 °C under 5% CO_2_ in the presence of 20 μM inhibitor in DMEM supplemented with 1% serum. For Chloroquine treatment, cells were washed in serum-free media and incubated for 24 h at 37 °C under 5% CO_2_ in the presence of 50 μM chloroquine in DMEM supplemented with 1% serum. For Rapamycin treatment, cells were washed in serum-free media and incubated for 6 h at 37 °C under 5% CO_2_ in the presence of 100 nM Rapamycin in DMEM supplemented with 1% serum. For Bafilomycin A1 treatment, cells were washed in serum-free media and incubated for 24 h at 37 °C under 5% CO_2_ in the presence of 100 nM Bafilomycin A1 in DMEM supplemented with 1% serum. The drug treatments were performed in low serum to make cells more sensitive to the treatment itself.

### 2.3. shRNA Interference

Short hairpin RNA sequences used were “CCGACTTGTTCCCTATGGAAAT” against Beclin1 (BECN1) involved in vesicle nucleation and “CCAACAGAACTTGATTGTAAATA” against autophagy related gene 16 like 1 (ATG16L1) involved in vesicle elongation. shRNA-GFP was used as scrambled.

One day before transfection, GT1 cells were plated on coverslips to reach 40–50% confluency at the time of transfection. GT1 cells were transfected with the shRNA by Lipofectamine LTX and PLUS Reagents (Invitrogen, Molecular Probes, Eugene, OR, USA, see manufacturer’s protocol). After 24 h from transfection, cells with integrated plasmid were selected by using Puromycin (Sigma-Aldrich, code p8833) 1 µg/mL in DMEM 10% FBS for 3 days before proceeding with immunofluorescence and immunoblot analysis.

### 2.4. RNA Extraction, cDNA Preparation and Real Time PCR

Total RNA was extracted from both untreated and NSC48478-treated GT1 cells using TriFast (EuroClone, see manufacturer’s protocol), following the manufacturer’s instructions. Next, RNA samples were quality-assessed running an RNA ScreenTape on the TapeStation system (both from Agilent Technologies, Santa Clara, CA, USA), in order to verify the integrity of the obtained RNA molecules.

Reverse transcription was carried out by using the High-Capacity RNA-to-cDNA Kit (Thermo Fisher Scientific, Waltham, MA, USA) starting from a 2 µg RNA/sample.

Specific primer pairs were designed for all the selected genes, i.e., *BECN1*, *ULK1*, *PIK3C3*, *DNM2*, *CDC42*, *PROM2*, *APPL1*, *SNX33*, *ARF1*, *VPS11*, *VPS18*, *CTSb*, and *GAPDH* used as normalizer, by using Primer3 web application (https://primer3.ut.ee, last accessed date 20 January 2022) with default parameters (Appendix A). When possible, primers overlapping exon-exon junctions were chosen in order to avoid possible genomic DNA amplification. Primer specificity was also tested by Primer-Blast tool (https://www.ncbi.nlm.nih.gov/tools/primer-blast/, last accessed date 20 January 2022).

Next, mRNA expression levels were evaluated using the Power SYBR Green PCR Master Mix (Life Technologies, Carlsbad, CA, USA). The RT-PCRs were carried out using 10 ng of cDNA/sample and 10 µM primers. The thermocycling conditions for RT-PCR were as follows: 95 °C for 10 min, followed by 40 cycles of 95 °C for 15 s and 59 °C for 1 min, plus the dissociation stage for the melting curve analysis, comprising 95 °C for 15 s, 60 °C for 15 s, and 95 °C for 15 s.

Each gene was analysed in triplicate, and GAPDH housekeeping gene was used as normalizer. The relative expression of each gene was calculated and normalized using the $2^{-\Delta \Delta Ct}$ method. *t*-Test was used to assess the presence of any significant value (*p* values: * *p* < 0.05; ** *p* < 0.01; *** *p* < 0.001).

### 2.5. Luciferase-Based Transcriptional Assay

One day before transfection, GT1 cells were cultured in 6-multiwell plates to reach 50% confluency at the time of transfection. Cells were transfected with Luciferase reporter vectors containing TFEB-responsive motifs (5xCLEAR) or Lamp1 promoter (500 ng for 6-well) using Lipofectamine LTX and PLUS Reagents (Invitrogen, Molecular Probes, Eugene, OR, USA, see manufacturer’s protocol). After 48 h from transfection, cells were lysated, and the luciferase activity was measured by a luminometer using the Dual-Glo Luciferase assay kit (Promega E2920). Data were quantified as previously published [18]. The luciferase activity values were normalized to the protein concentration of each sample. Each value represents the mean ± SEM of at least three independent experiments performed in triplicate.

### 2.6. Protein Extraction and Western Blotting

GT1 cells were washed twice with ice-cold phosphate-buffered saline (PBS) and total proteins were extracted in lysis buffer (25 mM Tris HCl pH 7.5, 150 mM NaCl, 5 mM EDTA, 1% TritonX-100) supplemented with protease inhibitor cocktail (Sigma Aldrich P8340) and phosphatase inhibitors (1 mM phenylmethylsulfonyl fluoride, 10 mM sodium fluoride and 1 mM sodium orthovanadate). The protein concentration was determined using the Bradford protein assay (Bio-Rad Laboratories, Inc., Hercules, CA, USA). The protein samples were boiled for 5 min in 5X Laemmli loading buffer, separated on SDS-PAGE, transferred onto polyvinylidene fluoride (PVDF) (GVS filter technology) and hybridized with the appropriate primary antibodies. The signal was detected using Enhanced Chemiluminescent Substrate method (Euroclone, EMP011005). For re-probing, PVDFs were stripped by incubation with 0.2 M NaOH for 5 min at 37 °C. Protein levels were quantified by densitometry using ImageJ software.

### 2.7. Plasmid Transfection

One day before transfection, GT1 cells were cultured on coverslips to reach 30–50% confluency at the time of transfection. GT1 cells were transfected with GFP-LC3 plasmid or TFEB-FLAG plasmid (250 ng for 24-well) using Lipofectamine LTX and PLUS Reagents (Invitrogen, Molecular Probes, Eugene, OR, USA, see manufacturer’s protocol). After 48 h from transfection, cells were fixed for indirect immunofluorescence.

### 2.8. Indirect Immunofluorescence and Confocal Microscopy

GT1 cells were cultured to 50–80% confluence in growth medium for 3 days on coverslips, washed twice in PBS, fixed in 4% paraformaldehyde (PFA) and permeabilized with 0.1% TritonX-100 for 30 min. Alternatively, for Rab protein staining, cells were fixed in TCA 10%-PBS at 4 °C for 15 min, washed twice in 30 mM Glycine-PBS and permeabilized with 0.2% TritonX-100 for 5 min. For TFEB-FLAG staining, cells were permeabilized with 0.075% saponin and 0.2% gelatine in PBS. For plasma membrane staining, cells were incubated for 10 min at 37 °C in the presence of 5 μg/mL WGA-Alexa 555 conjugate in Hanks’ balanced salt solution (HBSS) before proceeding with immunofluorescence. Fixed and permeabilized cells were processed for indirect immunofluorescence by incubating specific primary antibodies diluted in 5% BSA-PBS followed by incubation with fluorophore-conjugated secondary antibodies diluted in 5% BSA-PBS. Nuclei were stained by using DAPI (1:1000) in PBS.

For the LC3-positive dot count, we used ImageJ software and measures were obtained by analysing at least 25 cells/sample for at least three different experiments.

Pearson correlation coefficients (PCC) were measured as shown before [15]. The measure reflects a linear correlation between variables and how strong the relation is between them. The result has a value between −1 and 1, where the value −1 indicates a negative correlation, while the value +1 indicates a linear perfect correlation. For the analysis of TFEB nuclear localization, ImageJ software was used to measure the ratio of fluorescence intensity of nuclear TFEB with respect to the total cellular level of TFEB. The measures were obtained by analysing at least 30 cells/samples for at least three different experiments. Immunofluorescences were analysed by the confocal microscope LSM 700 Zeiss equipped with an oil immersion 63 × 1.4 NA Plan Apochromat objective, and a pinhole size of one Airy unit. Measurements of fluorescence intensity were taken on a minimum of three confocal stacks per condition from a single experiment (~25 cells), using LSM 700 Zeiss software ZEN. The background values raised by fluorescent secondary antibodies alone were subtracted from all samples.

### 2.9. Statistical Analysis

Each value represents the mean ± SEM of at least three independent experiments performed in triplicate. Indicated *p* values were obtained using the Student *t* test. Differences were considered statistically significant when * *p* < 0.05; ** *p* < 0.01; *** *p* < 0.001.

## 3. Results

Previous experiments revealed that inhibition of 37/67 kDa laminin receptor (LR, also known as RPSA) by naphthol-derived small molecules was able to control endocytosis and degradation of the receptor [19] and to reversibly affect the maturation of amyloid precursor protein (APP), acting through acidic compartments of mouse neuronal cells [15]. In view of the intimate correlation between the endocytic and autophagic pathways, we decided to assess the potential effects of this inhibitor on autophagy. To this end, we employed immunoblot analysis to monitor the abundance of LC3-II on extract of GT1 cells treated with NSC48478 inhibitor and grown in low serum medium (1% serum) in the presence or absence of Chloroquine (CQ), which was used to block the autophagosome turnover [20]. Indeed, the LC3-II amount, at a given time point, does not necessarily estimate the autophagic activity, because LC3-II itself is degraded by autophagy; thus, inhibition of autophagosome degradation as well as autophagy activation greatly increases the amount of LC3-II [21]. To correctly measure the autophagic flux, it is necessary to evaluate the amount of LC3-II delivered and degraded in lysosomes by using lysosome inhibitors [5]. As shown in Figure 1A, the analysis of the autophagic flux, obtained by evaluating LC3-II in low serum medium (control, lane 1) and upon NSC48478 treatment, showed an increased level of LC3-II upon inhibitor treatment (lane 2) that, as expected, was amplified when the cells were subjected to the lysosomal inhibitor CQ (lanes 3,4).

Data from densitometric analysis of LC3-II band in the immunoblotting assays (Figure 1A) were plotted, as shown in Figure 1B, and showed an increase of about 80% in the amount of LC3-II under NSC48478 administration compared with untreated conditions. The use of CQ with NSC48478 indicated a significant positive trend of autophagic flux induced by the 37/67 kDa LR inhibitor.

Additionally, the treatment with Rapamycin (Rapa), which is a known inducer of autophagy via the m-TOR pathway, increased LC3-II (Appendix A). In agreement with previous studies that demonstrated a very rapid and efficient clearance of newly formed autophagosomes in healthy neurons [22], the total level of LC3 significantly decreased under Rapa treatment (Appendix A), as compared with the untreated control (Appendix A), thus confirming that LC3 itself is a substrate of autophagy. When NSC48478 was used in combination with Rapa, LC3-II levels were higher compared to their levels with Rapa alone (Appendix A). These data suggest that NSC48478 is inducing LC3 lipidation and that if NSC48478 was acting through stimulation of canonical autophagy via the m-TOR pathway, when used in combination with Rapa, it would have determined a stronger decrease in LC3-II accumulation. Indeed, this is not the case.

Since Chloroquine, besides its known role in the inhibition of the canonical autophagic flux, is also able to induce activation of non-canonical autophagy and endolysosomal LC3 lipidation [23], we employed Bafilomycin A1 to check for LC3-II formation and further dissect the NSC48478 mechanism of action. Bafilomycin A1 is a specific inhibitor of vacuolar-type H(+)-ATPase, known to affect autophagosome–lysosome fusion inhibiting the canonical autophagic flux [24]. As expected, we found an increase in LC3 lipidation after Bafilomycin A1 administration to the cells (Appendix A). Interestingly, the combination of Bafilomycin A1 and NSC48478 induced a marked increase in lipidated LC3, as compared to untreated conditions and to the single treatments, respectively. This result strongly indicates that both Bafilomycin and NSC48478 contribute to LC3-II accumulation and suggests that NSC48478 induces LC3 lipidation additionally to the block of canonical autophagic flux by Bafilomycin, which to our knowledge, is not able to induce non-canonical autophagy as CQ instead does. These findings, together with data showing LC3-II accumulation under double CQ/NSC48478 treatment (Figure 1), indicate that we cannot exclude that CQ contributes to unconventional LC3 lipidation, but that in fact the increase in LC3-II is not exclusively due to the induction of non-canonical autophagy by CQ.

Additionally, we used immunofluorescence to examine the recruitment of LC3 on endosomes under CQ-alone conditions. As shown in Appendix A, the recruitment of LC3 on Rab5 positive endosomes, as well as on EEA1-positive endosomes, was negligible (PCC = 0.11 ± 0.02; PCC = 0.10 ± 0.03, respectively), as compared to control conditions.

To further confirm the ability of NSC48478 to stimulate the autophagic process, we evaluated the level of degradation of a known autophagic substrate, such as amyloid precursor protein (APP) [25,26]. As shown in Appendix A, the levels of APP were increased under CQ treatment, indicating that CQ functions as an inhibitor of lysosomal activity, while NSC48478 induces a decrease in APP levels, functioning as an inducer of lysosomal degradation (see discussion for details). These data strongly indicate that NSC48478 stimulates the biogenesis of the autophagic process rather than interfering with lysosomal activity.

Analysis of the number of autophagosomes represents another methodological approach that is routinely employed to evaluate the amplitude of autophagic activity [5].

The increase in the number of LC3 puncta in the presence of a lysosome inhibitor, compared to that in the absence of the inhibitor, represents the number of autophagosomes that would have been degraded during the treatment period. Therefore, by confocal microscopy analysis of immunofluorescence, we analysed the LC3-positive dots in transiently GFP-LC3 transfected cells grown under NSC48478 treatment and/or Chloroquine or Rapamycin (Figure 2A).

LC3-positive puncta were not increased under NSC48478 treatment (2.4 ± 0.82, expressed as mean ± SEM of LC3 puncta per cell), as compared to untreated conditions (2.3 ± 1.11), likely because of the degradation process. To test this possibility, we used CQ alone or together with NSC48478. As expected for normal autophagy-responsive cells [5], the use of CQ alone induced a marked increase in LC3 positive puncta (45 ± 7.4 dots/cell), as compared to untreated conditions (2.3 ± 1.11 dots/cell). Interestingly, the combined use of CQ and NSC48478 inhibitor highlighted the formation of ring-shaped structures, which were not observed in the other conditions.

The possibility that NSC48478 can interfere with autophagosomes degradation can be excluded due to the fact that we did not find any detectable modification, increase or larger puncta positive for LC3 under NSC48478 conditions (see Figure 2A, NSC48478 alone, panel).

Although Rapamycin alone or concomitantly used with NSC48478 induced LC3-II (Appendix A), it did not show peculiar structures positive for LC3, as compared to control conditions. In agreement with the immunoblot analysis (Figure 1 and Appendix A), and with previous studies about autophagy induction in neurons [22], it is likely that Rapamycin and NSC48478 alone or in combination did not induce autophagosomes accumulation because such structures are rapidly and efficiently degraded by lysosomes in healthy neurons.

Since it has been demonstrated that APP traffics along the endocytic pathway [26] and can partially be a substrate for autophagy degradation [25], by means of confocal microscopy, we analysed APP signal by a specific antibody [15]. As shown in the CQ+NSC48478 panel of the immunofluorescence assay (Figure 2B), the ring-shaped structures were positive not only for both APP and LC3 (see merged panels on the right), but also for the endocytic marker wheat germ agglutinin (WGA).

WGA binds GlcNAc- or sialic acid-containing oligosaccharides on glycosylated membrane proteins and is efficiently internalized into the endocytic pathway; thus, it is a good marker of the endocytic route [27]. Since WGA can access endosomes when internalized from cell surface, to characterize LC3-positive compartments, Alexa-555-conjugated WGA was internalized for 10 min at 37 °C, and GFP-LC3 transfected cells were then fixed, permeabilized, and processed with antibody against APP. There was extensive colocalization of internalized WGA with GFP-LC3 and APP in ring-shaped structures of cells treated with NSC48478+CQ (colocalization between WGA/LC3, PCC = 0.88 ± 0.01; APP/LC3, PCC = 0.89 ± 0.05). The colocalization of LC3-positive ring-shaped structures with WGA indicates a contribution by endocytic membranes to the formation of these structures.

Furthermore, the colocalization between LC3/WGA under Rapamycin treatment was highlighted (PCC = 0.88 ± 0.03) in dots that were different from the ring-shaped ones, which instead were evident under NSC48478+CQ administration, strengthening our hypothesis that NSC48478 acts through a different m-TOR independent pathway.

Growing evidence that members of the small GTPase Rab protein family, in addition to their role in intracellular vesicle trafficking, play a key role in the regulation of autophagy (reviewed in [28]), prompted us to employ different anti-Rab protein antibodies available in our lab to check levels and localization of the main Rab proteins involved in specific steps of the autophagic process. To this end, we analysed by confocal microscopy the distribution of the different Rabs (namely Rab5, Rab7, Rab8 and Rab27) and their colocalization with LC3 under NSC48478 treatment with/without CQ (Figure 3).

Specifically, we checked for the main represented Rab proteins previously reported to be involved in the biogenesis and/or maturation of the autophagic process: (i) Rab proteins involved in endocytosis (Rab5, early endosomes and Rab7, late endosomes) [29], (ii) Rab involved in transport of secretory and recycling vesicles towards plasma membrane (Rab8) [30], (iii) Rab known to be localized in the Golgi and to regulate exocytosis of vesicles in neuronal lines and synaptic vesicle release in neurons (Rab27) [31,32].

As shown in the CQ+NSC48478 panels of Figure 3, a strong colocalization was detected between Rab5 (panel d, PCC = 0.82 ± 0.02), Rab7 (panel e, PCC = 0.95 ± 0.05), and Rab8 (panel f, PCC = 0.88 ± 0.02) with LC3 puncta, respectively, while no colocalization between Rab27 and LC3 was revealed in these conditions (panel g, PCC = 0.12 ± 0.03).

Intriguingly, the analysis of anti-Rab immunoblotting on extracts from GT1 cells (Figure 4), showed that while the amount of Rab5, as well as Rab8, increased about 30% under NSC48478 conditions compared to untreated conditions, a marked decrease (about 85%) in the amount of Rab7 was induced by NSC48478 inhibitor. No variation in the amount of Rab27 was detected under NSC48478 treatment (bottom panel) (see discussion for details).

These data strongly suggest a contribution to autophagy by the Rab proteins regulating endocytosis/recycling of vesicles in neuronal cells, but not by Rab27, which typically regulates synaptic vesicle release from the Golgi apparatus. The origin of autophagosome membrane is still under debate; however, it is now clear that multiple sources can contribute to their formation [33], including endoplasmic reticulum (ER) [34], mitochondria [35] and plasma membrane [8,36]. Moreover, recent studies have demonstrated that LC3 lipidation is not restricted to double-membrane autophagosomes, but it can occur also on single-membrane macroendocytic vacuoles such as phagosomes, macropinosomes and entotic vacuoles, activating a noncanonical, m-TOR-independent autophagy pathway [6,7]. Furthermore, the employment of early endosome antigen-1 (EEA1) antibody in the immunoblotting on the cell extracts of GT1 cells grown in the same conditions as above (Appendix A) showed an increase in EEA1 under NSC48478 conditions and reinforced our hypothesis of an endocytic pathway contribution to autophagy.

Collectively, our results strongly suggest that the endolysosomal system contributes to the biogenesis of LC3-lipidated structures, and that, most likely, NSC48478 is acting through an m-TOR-independent mechanism.

To check this hypothesis, we analysed the phosphorylation of m-TOR (pm-TOR) at residue Ser-2448, by comparing the level of phosphorylated isoform to total m-TOR levels in cells treated or not with NSC48478, and/or Rapamycin. As shown in Figure 5, if on one hand Rapamycin, as expected, switched off m-TOR, inducing a decrease in its phosphorylation, on the other hand NSC48478 induced an increase in m-TOR phosphorylation levels, as compared to untreated conditions.

Collectively, the data presented so far suggest that the inhibitor acts in an opposite way to Rapamycin, that is, by blocking m-TOR-dependent autophagy and promoting an m-TOR-independent route (see discussion for details).

To test whether NSC48478 regulated the expression of autophagy genes, qPCR was performed to assess the mRNA levels of a group of three genes reported to be involved in different steps of the autophagic process (reviewed in [37]) and genes involved in the control of endocytic pathway [38] as well as in the maturation of autophagosomes [39]. Interestingly, some significant differences were highlighted between the inhibitor-treated and -untreated GT1 cells. Indeed, in agreement with our previously hypothesized independence from canonical autophagic pathway activation, *BECN1*, *ULK1* and *PI3KC3* expression levels were unchanged under inhibitor treatment (Figure 6A), thus suggesting that this pathway is not influenced by the NSC48478 inhibitor.

Conversely, as shown in Figure 6B, a significant difference was found in *DNM2* (Dynamin 2) [40], known to be a positive regulator of receptor-mediated endocytosis, and *CDC42* (Cell Division Cycle 42) [41], *APPL1* (Adaptor Protein, Phosphotyrosine Interacting With PH Domain And Leucine Zipper 1) [42], *SNX33* (Sortin Nexin 33) [43], and *ARF1* (ADP Ribosylation Factor 1) [44], known to be positive regulators of pinocytosis. These data were reinforced by the finding of unchanged levels of *PROM2*, a negative regulator of both pinocytosis and receptor-mediated endocytosis [41].

In addition, cells treated with the NSC48478 inhibitor showed a significant increase in *VPS11/18* and *CTSb*, genes known to be involved in the trafficking/fusion of vesicles and maturation of autophagosomes, respectively [39].

The transcription factor EB (TFEB) is the master regulator of genes involved in canonical autophagy, as well as lysosomal biogenesis and function, thus TFEB transcriptionally coordinates cellular degradative pathways [45]. Under nutrient-rich conditions, TFEB is mainly cytosolic and inactive, whereas under stressful conditions such as starvation or lysosomal dysfunction, it translocates to the nucleus to induce transcriptional upregulation of its target genes. Thus, to further characterize the molecular mechanism underlying NSC48478 activity, we analysed by confocal microscopy the nuclear localization of TFEB in transiently TFEB-FLAG-transfected GT1 cells, in the presence or absence of NSC48478 (Figure 7A).

Under inhibitor treatment, we observed an approx. 40% reduction in TFEB nuclear localization, as compared to the untreated conditions. This result, together with the immunoblot analysis of m-TOR phosphorylation (Figure 5), further supports the evidence that the inhibitor acts through a non-canonical autophagic pathway (see discussion for details).

Additionally, we assessed the functional activity of the TFEB by luciferase-based transcriptional assays. As shown in Figure 7B, upon NSC48478 treatment, we did not observe any significant variation in the activity of luciferase reporters under the control of TFEB.

Overall, these data suggest that NSC48478 treatment does not trigger the activation of TFEB-dependent transcriptional program. Hence, we can further conclude that our experiments ruled out the possibility of activation and/or influence of the gene expression program controlling lysosomal biogenesis and canonical autophagy.

Since the membrane origins of autophagosomes may involve multiple sources, in order to identify distinct ATG functional clusters, we started screening BECN1 and ATG16L1 involvement by knocking down with shRNAs. According to qPCR data where BECN1 levels were unchanged after inhibitor administration, BECN1 knockdown affected neither the formation of ring-shaped structures after treatment of cells with NSC48478 (Figure 8A,B), nor the lipidation of LC3 (Appendix A).

Interestingly, when we knocked down ATG16L1, which is known to regulate LC3 lipidation, both the formation of ring-shaped LC3-positive structures (Figure 8C,D) and the lipidation of LC3 were hampered (Appendix A).

Altogether, our data were supported and further strengthened by recent findings that other pharmacological modulators of autophagy can activate a non-canonical pathway driving ATG16L1-assisted LC3 lipidation [23,46].

## 4. Discussion

Our previous findings related to the regulation of APP maturation and intracellular trafficking through acidic compartments in neuronal cells and fibroblasts [15,47] prompted us to check for NSC48478 inhibitor involvement in the autophagy process. The resulting increased amount of LC3-II after 37/67 kDa LR inhibitor administration in mouse neuronal GT1 cells, allowed us to envisage the following *scenarios*: (a) if NSC48478 had inhibited degradation of autophagosome in lysosomes, we would have found more and/or larger LC3-positive autophagosomes and blocked autophagy substrate degradation (likely, this is not our case); (b) if NSC48478 had induced LC3 lipidation regulating biogenesis of autophagosomes, we would have found more and/or larger LC3-positive structures. Indeed, this was the case when we blocked lysosomal activity by using Chloroquine, which represents a well-established and standardized tool to test autophagic flux. However, as described by Jacquin et al. (2017) [23], Chloroquine, when used alone, is also able to induce endolysosomal LC3 unconventional lipidation and activation of non-canonical autophagy through a mechanism controlled by osmotic imbalance, which draws water into endolysosomal compartments, thus recruiting some autophagy proteins. Although we cannot completely rule out that CQ also activates a non-canonical autophagic process, the immunoblot analysis of LC3 using Bafilomycin A1 (which is known to inhibit autophagic flux without causing unconventional LC3 lipidation) in the presence or absence of NSC48478 (Appendix A) showed an accumulation of LC3-II; hence, this result strongly indicates that NSC48478 is able *per se* to induce LC3 lipidation. Moreover, the immunoblot analysis of APP (Appendix A) showed a decrease in APP levels in the presence of NSC48478, due to the fact that APP normally traffics through the endocytic pathway and can be a cargo of non-canonical LC3-assisted autophagy, reaching more efficiently the lysosomes for degradation, and thus sustaining the induction of an unconventional endosomal autophagic degradation by NSC48478. In agreement with our previous results showing how the phosphorylation of Akt was enhanced by 37/67 kDa LR inhibitor [15], we found that p-m-TOR levels increased after the use of NSC48478. These data converge towards a mechanism where NSC48478, through stimulation of Akt, affects the phosphorylation of m-TOR, inhibiting canonical autophagy while promoting a non-canonical pathway, where endocytic structures positive for different Rab proteins are involved and colocalize with LC3 (see schematic representation, Figure 9).

In addition to m-TOR phosphorylation, Akt has been demonstrated to inhibit canonical autophagy through BECN1 phosphorylation [48]. Additionally, in support of our results demonstrating m-TOR activation, besides the endocytic contribution to autophagic membrane biogenesis, a recent study also demonstrated that active m-TORC1 complex controls lysosomal biogenesis and endocytosis [49].

mTORC1 as well as ERK2 are the most characterized serine-threonine kinases responsible for TFEB phosphorylation, regulating its nuclear translocation and activity [45]. Under nutrient-rich conditions, m-TORC1 and ERK2 phosphorylate TFEB at residues Ser142 and Ser211, determining its cytoplasmic sequestration. Conversely, under stressful conditions such as nutrient deprivation or lysosomal deficiencies, TFEB is dephosphorylated and translocates into the nucleus, where it can bind to promoter motif in its target genes. According to immunoblot analysis that suggests an activation of the Akt-m-TOR signalling pathway, together with our previous results demonstrating that NSC48478 sustains ERK1/2 activation [15], we found that under inhibitor treatment, the percentage of nuclear TFEB was reduced as compared to the untreated condition. TFEB controls the expression of a gene network involved in both lysosomal biogenesis and canonical autophagy, namely the Coordinated Lysosomal Expression and Regulation (CLEAR) network, whose genes are characterized by a consensus sequence in their promoter regions specifically recognized by TFEB [39,50]. *BECN1*, *ULK1* and *PI3KC3* carry the CLEAR element in their promoter regions and are transcriptionally upregulated by TFEB. The qPCR analysis of these autophagic genes, together with the results of luciferase-based transcriptional assay, demonstrated that NSC48478 does not activate the canonical TFEB-dependent transcriptional program.

In this context, our findings of BECN1 exclusion from the mechanism of inhibitor functioning and ATG16L1 involvement in the formation of LC3-positive ring structures and LC3 lipidation further support that the inhibitor acts differently from canonical autophagy inducers. Non-canonical autophagy is characterized by independence from the upstream regulators responsible for initiation and nucleation of the phagophore in canonical autophagy, while it is dependent on the ubiquitin-like conjugation systems to coordinate the lipidation of LC3 [7,51]. In line with previous studies [9,10], we demonstrated that autophagy induction can occur independently from BECN1 activity; indeed, in BECN1-downregulated GT1 cells, NSC48478 induced the formation of LC3-positive ring-shaped structures and did not significantly alter the extent of LC3 lipidation, as compared to shRNA scrambled cells. He et al. in 2015 established a BECN1 knockout cell line, through which they demonstrated that the complete loss of BECN1 does not affect LC3 lipidation and autophagosomes formation; however, such autophagosomes are not functional and fail in the degradation of long-lived proteins [9]. Our experiments demonstrated that BECN1 activity is dispensable for LC3 lipidation induced by NSC48478; however, the efficiency of autophagic cargo degradation in BECN1-downregulated cells still needs further investigation. Unlike BECN1, our experiment highlighted a pivotal role of ATG16L1 in the lipidation of LC3 under NSC48478 treatment. Indeed, ATG16L1 downregulation in GT1 cells interferes with LC3-II lipidation and ring-shaped structure formation induced by the combined use of NSC48478 and CQ, as compared to scrambled shRNA transfected cells.

Previous studies have demonstrated that the WD repeat-containing C-terminal domain (WD40) of ATG16L1 is crucial for its recruitment to single membrane endolysosomal compartments and for LC3 lipidation during non-canonical autophagy, whereas canonical autophagy does not appear to be affected by deletion of the WD40 of ATG16L1 [52]. Deletion of the WD domain in ATG16L results in spontaneous AD pathology in mice and significant neuroinflammation, thus sustaining the importance of non-canonical autophagic proteins in the pathology of neurodegeneration and further strengthening the possibility to employ the regulation of non-canonical, as well as canonical autophagy, as a therapeutic strategy for AD [53]. In addition, the PI3KC3/BECN1 complex activity on early endosome membranes is regulated by Rab5 for the formation of autophagosomes [54]. Recent studies have further demonstrated that, under growth factor limitation, Rab5 activity is modulated by the catalytic subunit of the class IA phosphoinositide 3-kinase (PIK3CB/p110-β), positively regulating autophagosome biogenesis [55]. Our finding of Rab5 colocalization with LC3-positive ring-shaped structures, together with its marked increase after 37/67 kDa LR inhibitor treatment, suggests that Rab5 contributes to autophagy membrane biogenesis. In our experiments, the accumulation of Rab5 protein levels after blocking lysosomal function (NSC48478+CQ) indicates that Rab5 can be a substrate of lysosomal degradation.

Our results concerning qRT-PCR of APPL1 (adaptor protein pleckstrin homology domain, phosphotyrosine binding domain, and leucine zipper motif) overexpression under inhibitor treatment can be read in the light of previous reports, where it was described that APPL1 recruited to Rab5 complex on endosomes directly links APP-βCTF to Rab5 overactivation [56]. APPL1 stabilizes the GTP-active form of Rab5 on endosomes, slowing its switching to the GDP form and amplifying Rab5 signalling [57]. In this context, the abnormal recruitment of APPL1 to Rab5 endosomes is deleterious to endosome motility and endosomal cargo processing, including APP and synaptic plasticity.

Rab7 has a role in modulating the maturation of autophagosomes; indeed, it participates in microtubular transport of autophagosomes [58], as well as their fusion with late endosomes or lysosomes [59]. Our finding that the amount of Rab7 decreases upon NSC48478 treatment indicates that Rab7 can contribute to autophagic flux maturation and is a substrate of autolysosomal degradation, because its levels increase upon the blocking of lysosomal functionality by CQ.

Recent studies have cited Rab8 involvement in autophagy; indeed, Rab8 is implicated in a peculiar process of autophagy-based secretion of IL-1β [60], and it is involved in autophagosome maturation during autophagy clearance of microorganisms [61]. In our experiments, Rab8 showed the same behavior as Rab5; indeed, its levels increased under NSC48478 conditions and also accumulated after treatment both with CQ alone and in the presence of NSC48478. This finding suggests that Rab8, as well as Rab5, can contribute to autophagic flux. Non-canonical LC3-assisted phagocytosis is initiated at the plasma membrane, and it is neither dependent on components of the canonical autophagy pre-initiation complex nor is it subject to control by m-TOR [62,63].

On the other hand, the non-canonical pathway described by Niso-Santano [62] is different from the LAP because it demands a functional Golgi apparatus and is independent of any PI3KC3/BECN1 complex. Our results suggest that the NSC48478 inhibitor does not affect the ULK1/PI3KC3- BECN1 levels; thus, it likely acts independently from the pre-initiation complexes. Moreover, NSC48478 induces an increase in the phosphorylated Ser-2448 pm-TOR isoform. This result was expected, because we had previously shown activation of the Akt pathway [15], and previous reports had demonstrated that m-TOR is a direct substrate of Akt in the phosphorylation of Ser-2448 [16]. However, *VPS11/18* and *CTSb*, which are crucial in the late stages of autophagy, increased as a result. The first two genes have been reported to be involved in vesicular trafficking to allow the encounter between autophagosome and lysosomes, resulting in their fusion [64]. Moreover, the increased levels of *CTSb* upon NSC48478 treatment suggest that the inhibitor might influence the dynamics of both autophagosomal and lysosomal compartments [65]. These data are extremely important in light of the involvement of genome alterations in AD as well as the progression of other neurodegenerative diseases [66,67,68].

Of note, our previous finding of MAPK/ERK signalling inactivation induced by NSC48478 inhibitor [15] could be of great interest, if one considers the possibility that non-canonical autophagy could generate endolysosomal signalling hubs [69,70]. As such, this hypothesis might provide insight into the efficacy and function of 37/67 kDa LR inhibitor as an autophagy-modulating drug that possesses the property of endo-lysosomal lipidation of LC3.

## Figures and Tables

**Figure 1 cells-11-00466-f001:**
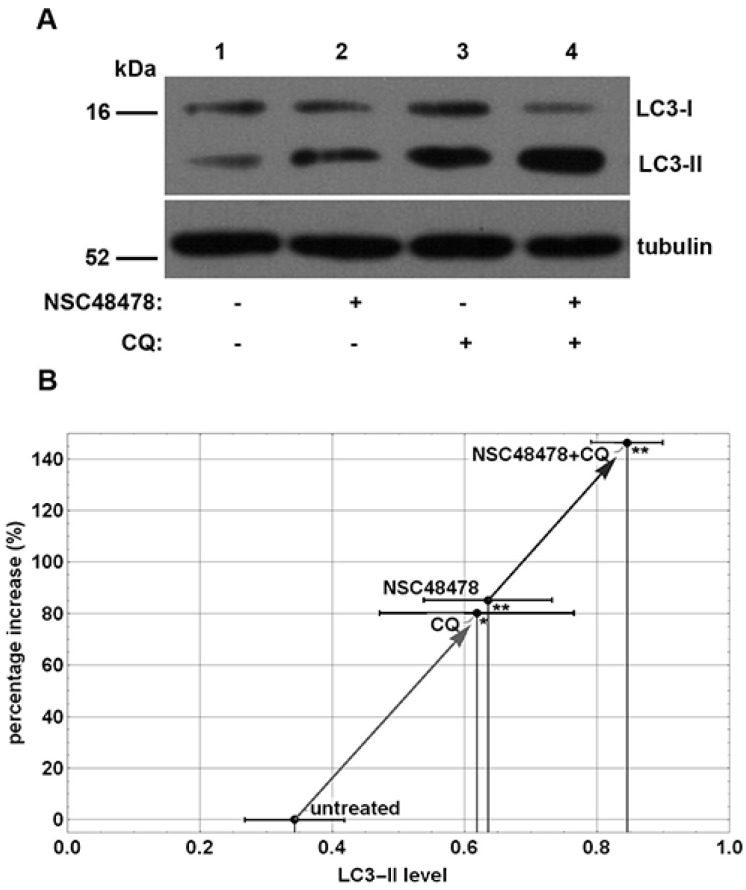
Inhibition of 37/67 kDa laminin receptor increases formation of lipidated LC3-II isoform. (**A**) GT1 cells were grown on dishes in 1% serum and left untreated (lane 1) or subjected to NSC48478 (lane 2), or/and CQ treatment (lanes 3,4). Treated (+) or not (−) samples were loaded on gels and analysed by SDS-PAGE and Western blotting with anti-LC3 antibody. Tubulin was used as loading and quantization control. The gels are representative of four independent experiments. (**B**) Plot represents the % increase in LC3-II isoform level in the different indicated conditions (quantified with respect to the total LC3 level and using tubulin as loading control). The arrows depict the trend of the % increase in LC3 induced by the inhibitor NSC48478. Mean values of LC3-II isoform level of four experiments were considered for representation. Data ± SEM are reported for each point. Variations in LC3 lipidation level in cells treated with NSC48478 and/or CQ were statistically significant as compared to the untreated control (* *p* < 0.05; ** *p* < 0.01).

**Figure 2 cells-11-00466-f002:**
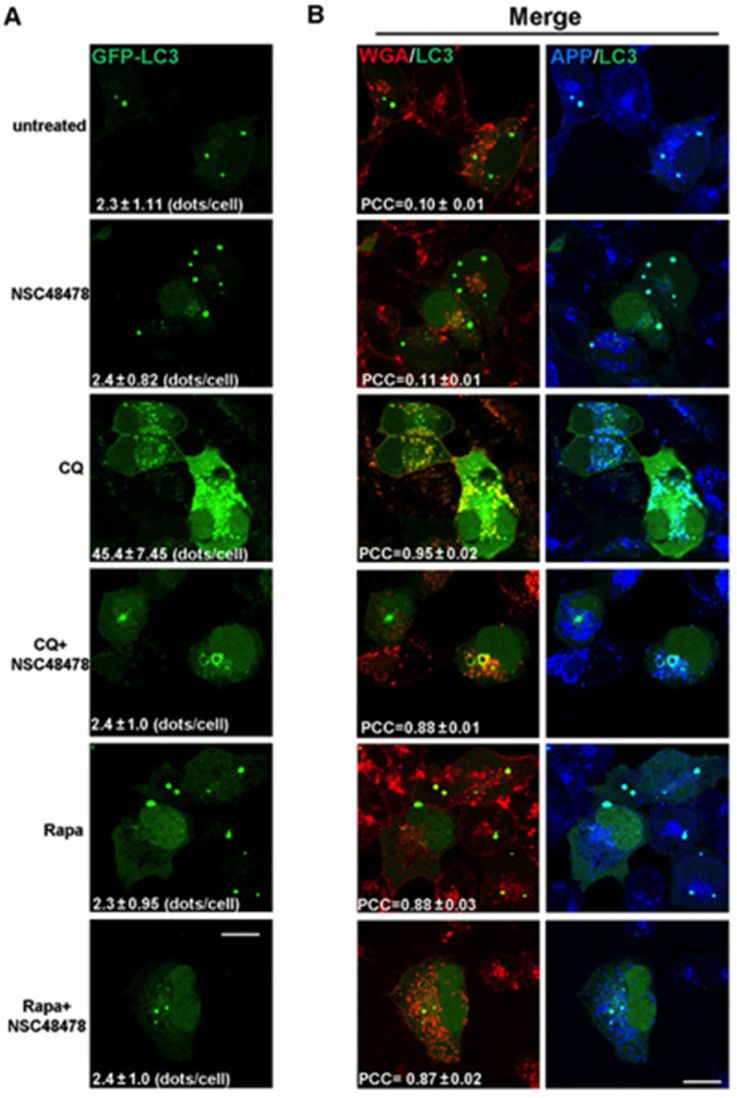
NSC48478 treatment induces formation of ring-shaped structures that are positive for LC3 and are of endocytic origin. (**A**) GT1 cells transiently transfected for GFP-LC3 were grown in low serum (1%) medium and left untreated (first upper panels), or treated as indicated. Cells were processed for immunofluorescence assay, and images were acquired by confocal microscopy. Panels show GFP-LC3 signal and number of LC3-positive dots/cell (±SEM). (**B**) The overlay (merge) between red/green and green/blue channels is shown on the right panels. Panels show the localization of WGA (Alexa-555 red), GFP-LC3 (green) and APP (Cy5, blue). Pearson’s correlation coefficient is an average value (N = 25) and based on at least four independent experiments. Scale bars: 10 µm; scale bars are shown on the bottom of figure and are the same for all panels.

**Figure 3 cells-11-00466-f003:**
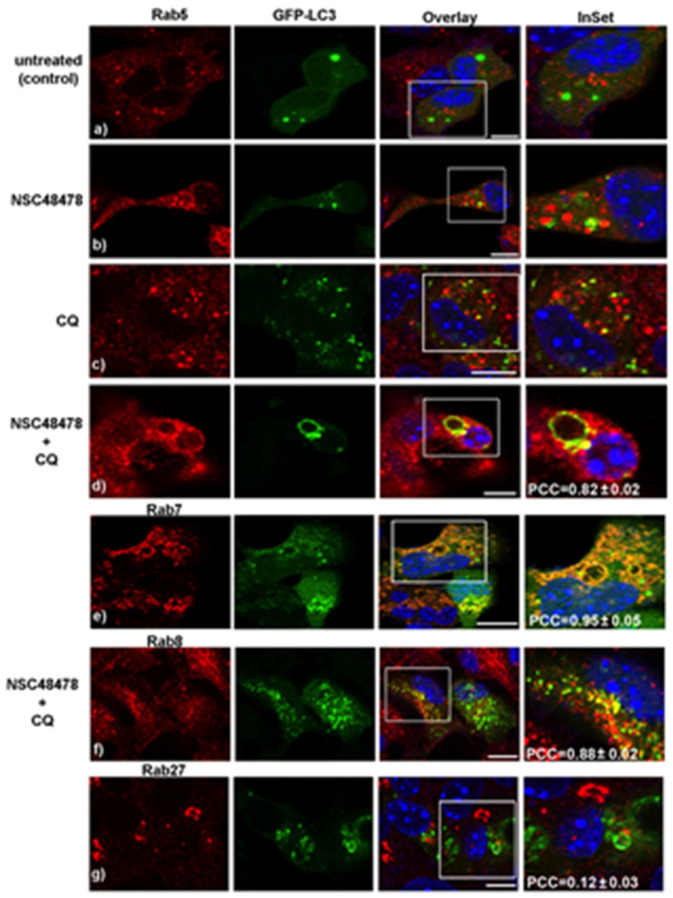
Rab proteins of the endocytic pathway, together with Rab of the recycling route, but not the secretory Rab27, participate in the formation of LC3-positive membranes. GT1 cells, treated as indicated, were processed for immunofluorescence as in Figure 2, with the exception that here, anti-Rab5 (panels **a**–**d**), anti-Rab7 (panel **e**), anti-Rab8 (panel **f**) and Rab27 (panel **g**) were used to measure the colocalization with GFP-LC3. PCC (±SEM) was calculated as described in methods. Scale bars: 10 µm.

**Figure 4 cells-11-00466-f004:**
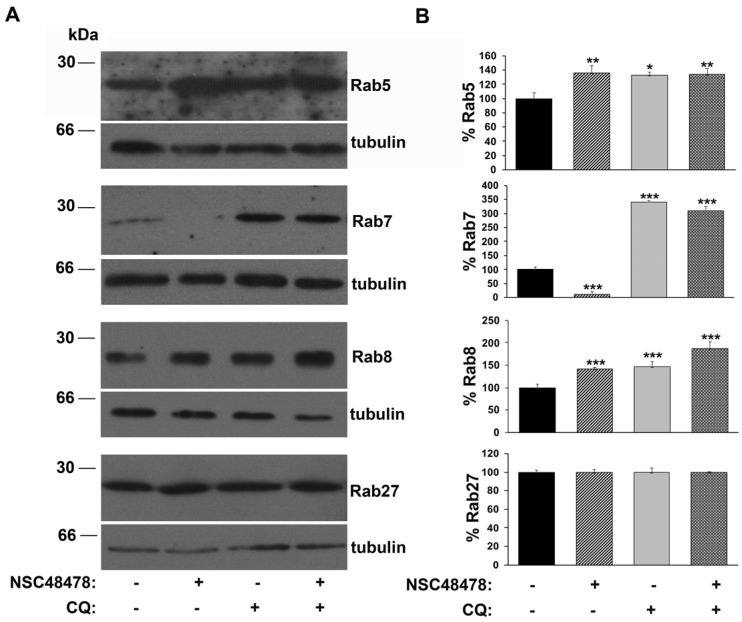
Inhibition of 37/67 kDa LR induces an increase in Rab5/Rab8 and a decrease in Rab7. (**A**) Cells were grown and treated as indicated in Figure 1. Western blotting analysis with the indicated anti-Rab antibodies was followed by anti-tubulin immunodetection, on the same membrane, for control of loading samples. Gels are representative of three independent experiments. (**B**) Data were obtained by imposing as 100% the Rab levels under untreated conditions. To note: no variation in Rab27 levels was detected (* *p* < 0.05; ** *p* < 0.01; *** *p* < 0.001).

**Figure 5 cells-11-00466-f005:**
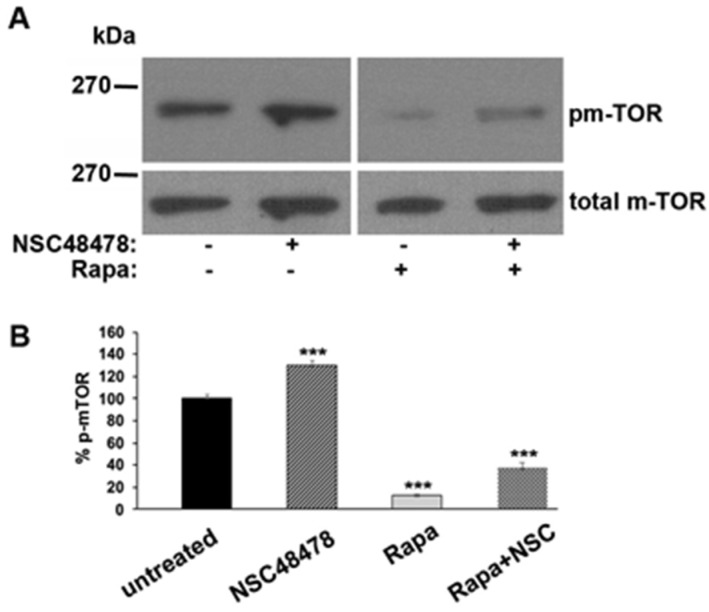
Phosphorylated m-TOR Ser-2448 isoform increases after NSC48478 administration. (**A**) GT1 cells grown on dishes in 1% serum were left untreated (−) or treated (+) as indicated. Rapamycin 100 nM was used as control for the procedure. SDS-PAGE followed by immunoblotting with anti-phospho Ser-2448 antibody revealed an increase in p-m-TOR after NSC48478 administration. Total m-TOR was revealed by probing the same membranes with anti-m-TOR antibody after stripping. (**B**) The gels are representative of three independent experiments plotted in the graph, where NSC48478 and Rapa treatments were compared to untreated conditions (expressed as 100%) (*** *p* < 0.001).

**Figure 6 cells-11-00466-f006:**
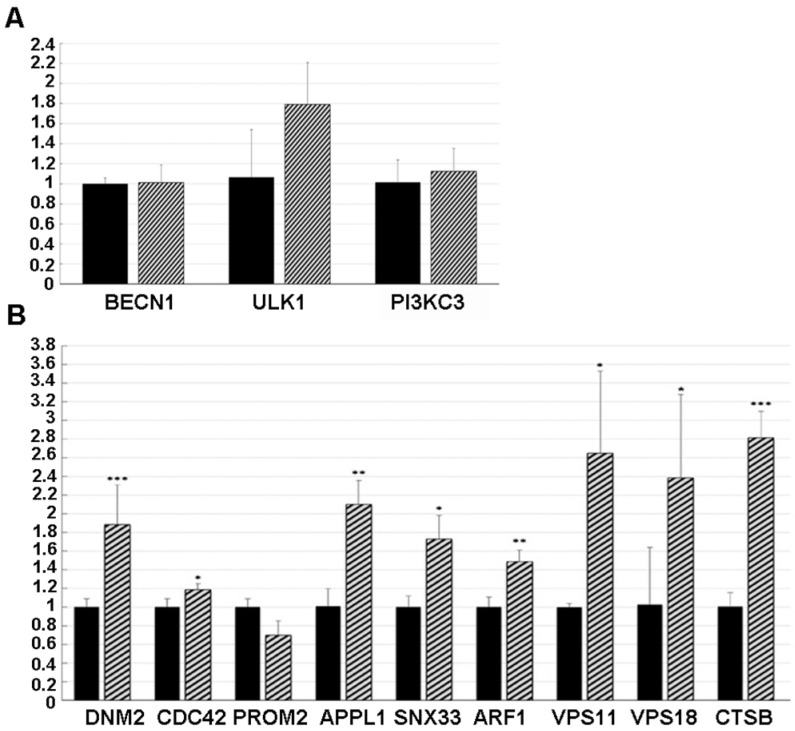
Gene expression levels as evaluated in both untreated (black histograms) and NSC48478 treated (line histograms) GT1 cells, by qPCRs. All reactions were performed in triplicate and normalized by using GAPDH as internal reference. (**A**) Canonical-autophagy related genes did not differ between the tested conditions. (**B**) Both endocytosis-related and autophagosome maturation-associated genes show some significant results. In particular, *DNM2*, *CDC42*, *APPL1*, *SNX33*, *ARF1*, *VPS11/18*, and *CTSb* were all significantly more expressed in the treated cells (* *p* < 0.05; ** *p* < 0.01; *** *p* < 0.001).

**Figure 7 cells-11-00466-f007:**
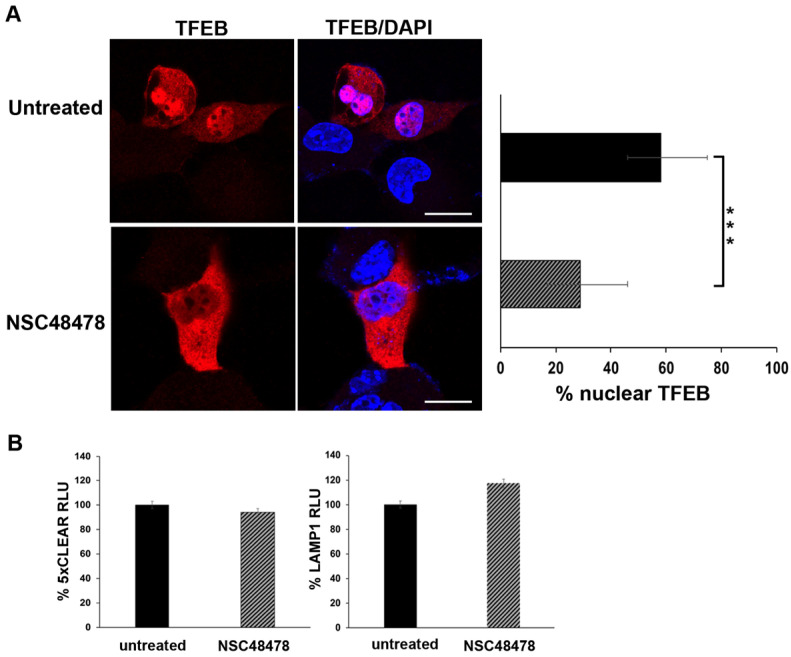
TFEB nuclear localization is affected by receptor inhibition. (**A**) GT1 cells were transiently transfected with TFEB-FLAG and left untreated or treated with NSC48478. Cells were then processed for immunofluorescence analysis (see methods for details). Nuclei were stained with DAPI. The histogram shows the percentage of TFEB nuclear accumulation, which was calculated by the ratio of fluorescence intensity of nuclear TFEB to the total level. The measures are the mean value ± SEM obtained by analysing at least 30 cells/samples for at least three different experiments (*** *p* < 0.001). Scale bars: 10 μm. (**B**) The cells were transiently transfected with the luciferase vectors containing 5 TFEB-responsive elements (5xCLEAR) or Lamp1 promoter. Luciferase activity reported on the histograms is shown as % RLU (relative luciferase activity) in both untreated or NSC48478-treated cells. Mean values were obtained by three independent experiments, and variations were not statistically significant.

**Figure 8 cells-11-00466-f008:**
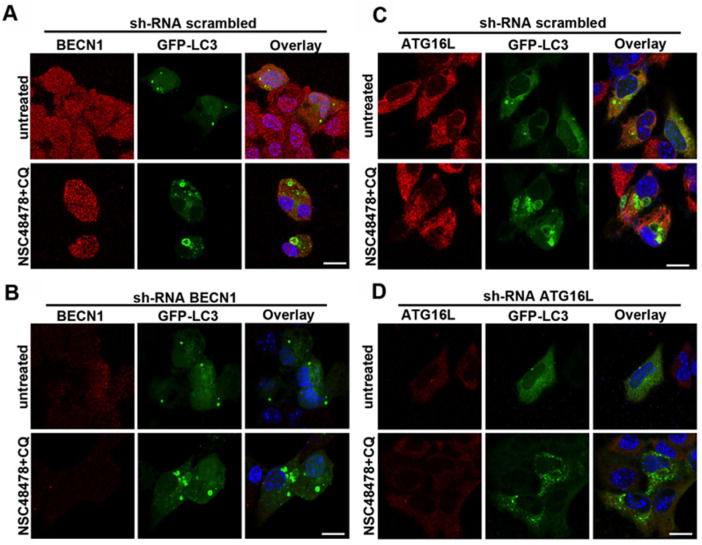
Effects of BECN1 and ATG16L1 knockdown on LC3-positive ring structures. GT1 cells were transiently transfected with specific shRNAs for BECN1 (**A**,**B**) or ATG16L1 knock down (**C**,**D**) and with GFP-LC3 (see methods for details). Cells were treated or not with NSC48478+CQ and processed for immunofluorescence analysis. Note the absence of LC3-positive ring structures in shRNA ATG16L1 (panel **D**). ShRNA-GFP were used as scrambled (panels **A**,**C**).

**Figure 9 cells-11-00466-f009:**
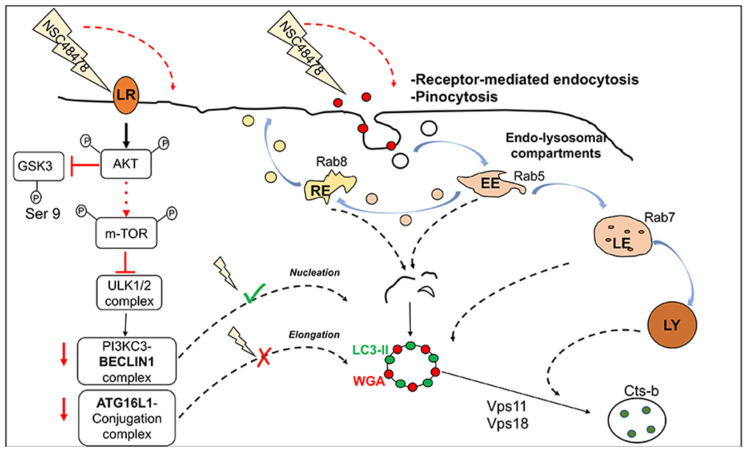
The 37/67 kDa laminin receptor inhibitor, NSC48478, induces the activation of non-canonical m-TOR-independent autophagic pathway. Left part of the scheme: upon NSC48478 administration, we previously demonstrated activation of Akt and GSK3b switch-off. Here, we demonstrate that inhibitor treatment induces phosphorylation of m-TOR (most likely through Akt activation) and does not affect the preforming autophagosome complex ULK1/PI3KC3/BECN1. Upon NSC48478 treatment, knockdown of BECN1 (red arrow) does not affect LC3-positive structures; while knockdown of ATG16L1 (red arrow) affects both formation of LC3 structures and LC3-II. Right part of the scheme: NSC48478 administration induces formation of endosomal structures positive for both LC3 and endocytic marker WGA, together with contribution of Rab 5-7-8. qRT-PCR analysis shows a significant upregulation of *Vps11*, *Vps18*, and *CTSb*, indicative of an autophagy event. LR: laminin receptor; RE: recycling endosomes; LE: late endosomes; Ly: lysosomes.

## Data Availability

Not applicable.

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
