# Peer review of "Activation of Non-Canonical Autophagic Pathway through Inhibition of Non-Integrin Laminin Receptor in Neuronal Cells"

_cells, 2022, doi:10.3390/cells11030466_

Round 1
Reviewer 1 Report
Autophagy is a lysosomal degradation pathway that has been associated with neurodegenerative disorders such as AD. In this manuscript, the authors investigate the role of NSC48478, a laminin receptor inhibitor, in autophagy. They found that NSC48478 induces LC3 conversion and autophagy activation through a non-canonical mTOR-independent endocytic pathway, thus suggesting a potential therapeutic function of NSC48478 in neurodegeneration-related pathological conditions. Overall, the molecular mechanism of NSC48478 in autophagy is still very obscure and the results are too preliminary to reach the author’s conclusion. I am afraid that I cannot support for publication at current stage.
Major:
- Knockdown of knockout core ATG proteins will be important to demonstrate that NSC48478 acts via a non-canonical autophagy pathway.
- The formation ring-shaped structures in cells treated with CQ and NSC48478 could be due to excessive endocytic trafficking or membrane fusion. It will also be important to determine how NSC48478 regulates Rab function during autophagic process.
Author Response
Response to Reviewer 1 comments
Reviewer 1. Autophagy is a lysosomal degradation pathway that has been associated with neurodegenerative disorders such as AD. In this manuscript, the authors investigate the role of NSC48478, a laminin receptor inhibitor, in autophagy. They found that NSC48478 induces LC3 conversion and autophagy activation through a non-canonical mTOR-independent endocytic pathway, thus suggesting a potential therapeutic function of NSC48478 in neurodegeneration-related pathological conditions. Overall, the molecular mechanism of NSC48478 in autophagy is still very obscure and the results are too preliminary to reach the author’s conclusion. I am afraid that I cannot support for publication at current stage.
Major:
- Knockdown of knockout core ATG proteins will be important to demonstrate that NSC48478 acts via a non-canonical autophagy pathway.
Response: We strongly agree with the referee’s observation. Thus, as suggested, we have performed knock down of two different ATG genes that are in distinct ATG functional clusters: BECN1 and ATG16L1. We performed the main experiments related to the analysis of both LC3-positive ring-shaped structures and LC3 lipidation (New Figure 8 and Figure S4) in different cell culture conditions (NSC48478 treatment compared to untreated controls). As reported in the revised version of the manuscript (pags 2 introduction, pag 13-14 results and pags 15-16 discussion) and as shown in the new Figures 8 and S4, we found that while BECN1 knock down did not affect the features described above, ATG16L1 knock down resulted both in the absence of LC3-ring structures and in low LC3-II production, as compared to control cells. These data, strengthened by recent findings of ATG16L1 function in the non-canonical autophagy (Wang et al, EMBO J 2021), strongly suggest that the receptor inhibition is driving a non-canonical pathway where ATG16L1 is involved. Moreover, the exclusion of BECN1 from NSC48478 inhibitor functioning is in agreement with the already shown data (Figure 6), where by qPCR analysis we did not found any variation in BECN1 levels after inhibitor treatment.
- The formation ring-shaped structures in cells treated with CQ and NSC48478 could be due to excessive endocytic trafficking or membrane fusion. It will also be important to determine how NSC48478 regulates Rab function during autophagic process.
Response: Our hypothesis is that NSC48478 does not directly control Rab activation/functioning, rather it is likely that the receptor inhibition causes modulation of downstream signaling pathway that activates non-canonical m-TOR-independent autophagy. The involvement of different Rabs demonstrates the contribution of the endolysosomal compartments to membrane origin of LC3-positive structures, whose formation is induced by non-canonical autophagy pathway (see revised version of the manuscript).

Reviewer 2 Report
The authors provide the characterization of the activation of a non-canonical autophagy pathway via the inhibition of non-integrin laminin receptor in a neuronal immortalized cell line. The experimental approach includes imaging experiments upon various treatment able to impact on the autophagy pathway to evaluate the flux and a set of experiments that evaluate protein activation and/or the impact on gene expression upon the non-integrin laminin receptor inhibition.
Some points need to be addressed before publication.
Line 141: why was the experiment performed in low serum (1%)?
Figure 1 shows the western blot quantification as the average of 4 independent experiments, but no SD or SEM are provided. To allow readers to have a clear idea of the variability among experiments I would add the standard deviation or the standard error of the mean to panel B. Does the sentence “All data were statistically significant” in the figure legends mean that all the differences were significant? If so, what is the p values and what are the stastistical tests that were used?
Fig.2: the error value for the number of dots per cells needs to be matched in term of significative figures after the point to the average. Example: 2.3±1.1 in the first image.
If PCC is an avarage value, would it be possible to provide the SD or SEM, to evaluate the variability among different images? This would be needed for both Fig. 2 and Fig. 3
It would be interesting to see if what is the effect of the NSC48478 inhibitor at the protein level on the genes that where evaluated via qPCR. Another interesting experiment would be to the study of the translocation of TFEB transcription factor to the nucleus to further clarify the actual pathway by which autophagy is upregulated via NSC48478.
Minor:
- I have mainly seen mTOR written and “mTOR” rather than “m-TOR”, so I would suggest to change it.
- Check for typos and English
Author Response
Response to Reviewer 2 comments
Reviewer 2. The authors provide the characterization of the activation of a non-canonical autophagy pathway via the inhibition of non-integrin laminin receptor in a neuronal immortalized cell line. The experimental approach includes imaging experiments upon various treatment able to impact on the autophagy pathway to evaluate the flux and a set of experiments that evaluate protein activation and/or the impact on gene expression upon the non-integrin laminin receptor inhibition.
Some points need to be addressed before publication.
Point 1: Line 141: why was the experiment performed in low serum (1%)?
Response: Thanks to our previous published data (see Bhattacharya et al., JPM 2020 and IJMS 2020) we established that all the treatments of cultured cells would have be performed in low serum (1%) to make cells more sensitive to the inhibitor treatment itself (see materials and methods subheading “cell culture and drug treatment”). Thus, to maintain the protocol conditions, where we know the inhibitor is functioning, we determined the autophagic flux under low 1% serum only (control conditions, starvation where LC3-II is formed, Figure 1A), and different treatment with CQ and Rapamycin, maintaining 1% serum in the cell culture media.
Point 2: Figure 1 shows the western blot quantification as the average of 4 independent experiments, but no SD or SEM are provided. To allow readers to have a clear idea of the variability among experiments I would add the standard deviation or the standard error of the mean to panel B. Does the sentence “All data were statistically significant” in the figure legends mean that all the differences were significant? If so, what is the p values and what are the stastistical tests that were used?
Response: We strongly agree with the referee’s comment. Indeed, we have now reported in the main Figure 1 of the revised manuscript, the SEM of the mean from four independent experiments. The statistical significance is referred to the increase in LC3 lipidation in cells treated with NSC48478 or/and CQ as compared to the untreated control. Statistical analysis was already reported in the materials and methods section (p values were obtained using the Student t test, pag 5, line 245). Furthermore, p values have been now reported in the figure legend 1, too (pag 6, line 294-296 of the revised version).
Point 3: Fig.2: the error value for the number of dots per cells needs to be matched in term of significative figures after the point to the average. Example: 2.3±1.1 in the first image.
Response: We apologize for lacking the error value in the dots number. We remedied this fault by introducing the exact value in term of significative figure. Please, refer to new Figure 2 (pag 7).
Point 4: If PCC is an average value, would it be possible to provide the SD or SEM, to evaluate the variability among different images? This would be needed for both Fig. 2 and Fig. 3.
Response: We agree with this comment and reported the SEM values in the revised Figures 2 and 3 (pags 7-9).
Point 5: It would be interesting to see if what is the effect of the NSC48478 inhibitor at the protein level on the genes that where evaluated via qPCR. Another interesting experiment would be to the study of the translocation of TFEB transcription factor to the nucleus to further clarify the actual pathway by which autophagy is upregulated via NSC48478.
Response: We strongly agree with the referee’s comment. We have performed two new experiments to check for TFEB localization and transcriptional activity after administration of NSC48478, as compared to control cells (pags 12-13 of the revised version and discussion): “The transcription factor EB (TFEB) is the master regulator of genes involved in canonical autophagy, as well as lysosomal biogenesis and function, thus TFEB transcriptionally coordinates cellular degradative pathways. In resting conditions, TFEB is mainly cytosolic and inactive, whereas under stressful condition such as starvation or lysosomal dysfunction, it translocate to the nucleus to induce transcriptional upregulation of its target genes. Thus, to further characterize the molecular mechanism underlying NSC48478 activity, we analysed by confocal microscopy the nuclear localization of TFEB, in transiently TFEB-FLAG-transfected cells, in the presence or not of NSC48478 (new Figure 7A). Under inhibitor treatment, we found about 40% reduction in TFEB nuclear localization, as compared to the untreated conditions. This result, together with the immunoblot analysis of m-TOR phosphorylation (Figure 5), further sustains the evidence that the inhibitor is acting through a non-canonical autophagic pathway (see discussion for details). Additionally, we assessed the functional activity of the TFEB by luciferase-based transcriptional assays. As shown in Figure 7B, upon NSC48478 treatment, we did not observe any significant variation in the activity of luciferase reporters under the control of TFEB. Overall, these data suggest that NSC48478 treatment does not trigger the activation of TFEB-dependent transcriptional program. Hence, we can further conclude that our experiments rule out the possibility of activation and/or influence the gene expression program controlling the lysosomal biogenesis and canonical autophagy”.
It would be interesting to see the effect of the NSC48478 inhibitor at the protein level on the genes that where evaluated via qPCR. We are working for producing these data that will be included in another soon planned manuscript.
Minor:
- I have mainly seen mTOR written and “mTOR” rather than “m-TOR”, so I would suggest to change it.
- Response: done.
- Check for typos and English
- Response: done.
